# Temporal Phosphorus Dynamics in Shallow Eutrophic Lake Suwa, Japan

**Yutaka Ichikawa [1],\*, Takashi Kunito [2] and Yuichi Miyabara [3,4]**

1    Department of Mountain and Environmental Science, Interdisciplinary Graduate School of Science and Technology, Shinshu University, 5-2-4 Kogandoori, Suwa 392-0027, Nagano, Japan

2    Department of Environmental Science, Faculty of Science, Shinshu University, 3-1-1 Asahi, Matsumoto 390-8621, Nagano, Japan; kunito@shinshu-u.ac.jp

3    Education and Research Center for Inland Water and Highland Environment, Faculty of Science, Shinshu University, 5-2-4 Kogandoori, Suwa 392-0027, Nagano, Japan; miyabar@shinshu-u.ac.jp

4    Institute of Mountain Science, Shinshu University, 5-2-4 Kogandoori, Suwa 392-0027, Nagano, Japan

\*    Correspondence: itikawa.lake16@gmail.com; Tel.: +81-266-52-1955

**Abstract:** It can be difficult to decrease the water phosphorus (P) concentration in eutrophic shallow lakes, even if the external P loading is reduced, owing to a high level of internal P loading to surface water from sediment. However, in shallow Lake Suwa, Japan, lake water P concentration has largely decreased in recent years owing to low levels of internal P loading, as well as declining external P loading. We measured water/sediment P and iron (Fe) concentrations and the P release rate from sediment in Lake Suwa, and then compared it with data from the 1970s. In the 1970s, the P concentration throughout the lake water was high during the hypoxic period. Recently, however, the P concentration has increased only in the hypolimnion during the hypoxic period. This suggests that internal P loading from sediment to surface water has largely been suppressed during the hypoxic period in recent years. This may be due to (i) stronger water stratification from global warming, (ii) a greater decrease in the P release rate from the sediment owing to a decline in sediment P concentration from the 1970s to 2020, and (iii) stronger formation of the Fe–P cycle in Lake Suwa recently, compared with that in the 1970s. Our results indicated the need to reduce both external P loading, and internal P loading from sediment to water, for effective water quality improvement in shallow lakes.

**Keywords:** phosphorus; iron; water quality in shallow lake; sediment; Fe–P cycle; global warming

## 1. Introduction

Increased nutrient inflows from lake catchments caused by anthropogenic activity have caused eutrophication in lakes around the world [1]. In a relatively short period this has resulted in large changes in lake ecosystems, such as high levels of phytoplankton production due to increased phosphorus (P) concentration, which is a limiting nutrient for primary production in many lakes [2]. This high phytoplankton production may have severe impacts on both the lake environment and human health because, for example, some *Microcystis* spp. pose health hazards to various organisms, including humans, due to their toxic products [3,4]. For these reasons, reduction in external P loading has been attempted in various lakes to decrease water P concentration. However, in shallow lakes, a reduction in external P loading does not easily decrease the lake water P concentration because of the influence of internal P loading [5,6]. Internal P loading is a path through which P accumulated in the lake sediment is released into lake water, and these values can be considerable for shallow lakes [1]. For example, in Lake Kasumigaura, Japan (average water depth 3.8 m [7]), although external P loading had decreased from 179 mg m$^{-2}$ year$^{-1}$ in 1990 to 139 mg m$^{-2}$ year$^{-1}$ in 2020 as sewerage development progressed, the total P (TP) concentration in the lake water had increased from 61 μg L$^{-1}$ to 92 μg L$^{-1}$ [8,9]. In Lake Søbygård, Denmark (average water depth 1.2 m [10]), external P loading per catchment

had decreased drastically from 28–33 mg m$^{-2}$ year$^{-1}$ (1978–1981) to 2–7 mg m$^{-2}$ year$^{-1}$ (1984–1995), but the TP concentration in the lake water during the summer had only decreased slightly from 900–1600 µg L$^{-1}$ to 400–1000 µg L$^{-1}$ [11].

In Lake Suwa, Japan, increased anthropogenic nutrient inflow caused severe eutrophication during the 1960s and 1970s, resulting in phytoplankton blooms in summer, mainly of Cyanophyta [12]. In addition, in the 1970s, high levels of internal P loading, P release from sediments driven by high water temperature, and hypoxia, increased the P concentration throughout the lake water during the summer. Also, the high levels of P release from the sediment caused phytoplankton blooms through increased levels of highly bioavailable reactive phosphate (RP) in the water. However, with the development of sewage systems in the catchment from 1979 [13], the external P loading decreased from approximately 300 kg day$^{-1}$ in the 1970s to less than 100 kg day$^{-1}$ in the 2000s, and values have also decreased gradually in recent years [8,14,15]. With respect to internal P loading, the P release rate in the 1970s was estimated to be approximately 70 kg day$^{-1}$ [16], but since the 1980s there have been few studies of internal P loading from sediments. Accompanying the decreased external P loading, the TP concentration in the lake water was greatly reduced, from 310 µg L$^{-1}$ in the late 1970s (1977–1979) to 50 µg L$^{-1}$ in the early 2000s [8,17], and has remained at low levels of around 50 µg L$^{-1}$ since 2000 [18]. According to Miyabara [18], despite hypoxia in the lake bottom layer during the summer, there has not been a clear increase in the lake water P concentration during summer since the 2000s, in contrast to values found in the 1970s to 1980s. These results suggest that a reduction in the lake water P concentration has been successfully accomplished in Lake Suwa by reducing the external P loading, although Lake Suwa is shallow and thus the internal P loading seems to be substantial.

Clarifying the mechanism that prevents a summer increase in the P concentration in this shallow lake, despite the formation of hypoxia, can provide useful information to enable the improvement of water quality in other shallow lakes. To uncover the mechanism of water quality improvement in Lake Suwa, we compared the P concentration in the lake water and sediment, the P forms in the sediment, and the P release rate between the 1970s and in recent years (2019–2022).

## 2. Materials and Methods

### 2.1. Study Site and Samples

Lake Suwa is situated in the center of Nagano Prefecture, Japan, and has a total area of 13.3 km$^2$. The mean depth of the lake is approximately 4 m, and it has a catchment area of 531 km$^2$. The water storage capacity is approximately 62,987 × 10$^3$ m$^3$, and the average residence time is approximately 46 days [19]. From March to December 2019, the water temperature was approximately 5–20 °C during the circulation period and 20–27 °C at 0 m during the stratification period (unpublished results). The average annual precipitation is 1301.5 mm [20], and there are several hot springs in the catchment area [21].

We measured dissolved oxygen (DO) concentration with a HACH HQ30d DO meter (TOA DKK, Tokyo, Japan) approximately every two weeks from April to December 2019 at a depth of 0 m (epilimnion), 5 m (hypolimnion), and just above the bottom (5.5 m). The oxidation–reduction potential (ORP) was measured with a HM-31P ORP meter (TOA DKK, Tokyo, Japan) from May to December 2019. Water temperatures at depths of 0.5 m and 5.0 m were measured with an installed DO/water temperature logger (HOBO Dissolved Oxygen Logger, Onset, Bourne, MA, USA) which was fixed in the lake, every 10 min from April to December 2019. Water density was calculated from the measured water temperature according to Kell et al. [22].

Water samples from the lake surface (0 m) and near the bottom (5 m) were collected at the center of the lake with a Van Dorn water sampler (Rigo, Tokyo, Japan) twice a month from April to December 2019. A part of the water sample was filtered through a Whatman GF/C filter (Cytiva, Marlborough, MA, USA) to separate the suspended

particles. Unfiltered water was used for TP and total iron (Fe) (TFe) measurements as described below.

Sediment cores were collected at the center of the lake every month from May to December 2019 and in August 2020 using acrylic tubes (inner diameter 4.0 cm) with a core sampler (Rigo, Tokyo, Japan). After removing the overlying water via siphon, sediment samples were extruded, and one layer (0–4 cm, in 2019) or four layers (0–2 cm, 2–4 cm, 4–6 cm, 6–8 cm, in 2020) were sliced out, followed by centrifugation (3000 rpm, 15 min) to obtain pore water. Pore water was obtained after filtration through a 1 μm syringe filter for chemical analyses. Settling particles were collected from May to December 2019 with sediment traps at intervals of 1–2 months. The sediment traps consisted of bundles of three PVC pipes (10 cm inner diameter and length 50 cm), set at depths of 3 m and 5 m. Sediments and settling particles were freeze-dried for chemical analyses. Sediment samples collected with a core sampler in 1977 in Lake Suwa, which had been stored air-dried, were also used for chemical measurements.

### 2.2. Chemical Analyses

RP in the filtrated water sample was measured via the molybdenum blue method [23]. TP in non-filtrated water and dissolved total phosphorus (DTP) in filtrated water were measured after autoclave persulfate digestion [24].

For Fe measurement in sediment and settling particles, the samples were treated using autoclave persulfate digestion [24]. The Fe concentration was analyzed with a flame atomic absorption spectrometer AA-630-12 (Shimadzu, Kyoto, Japan). Phosphorous in sediment and settling particles was sequentially extracted with citrate-dithionate-bicarbonate (CDB) for 30 min, 1 M NaOH for 16 h, and 2 M HCl for 30 min [25]. Inorganic P (Pi) in these fractions is regarded as Pi bound with redox-sensitive Fe (CDB-Pi), P bound with Al (NaOH-Pi), and P bound with Ca (HCl-Pi). Pi in the fractions was measured via the molybdenum blue method, and organic P was estimated by subtracting Pi from TP in each fraction. The sum of the TP in these three fractions was treated as the TP value of the sediment and settling particles. The sum of organic P content in the three fractions was denoted as Org-P. Fe in the CDB fraction was regarded as reactive (redox-sensitive) Fe. These concentrations in sediment and settling particle samples were expressed as a mean $\pm$ standard error on a dry weight basis.

### 2.3. Phosphorus Release Test

A P release test was conducted using sediment samples in 15 acrylic tubes (inner diameter, 4.0 cm) in a dark room at a temperature of 18 °C; the height of the sediment was adjusted to 15 cm and 350 mL of filtered lake water was poured above the sediment. During the experiment, to keep the DO concentration below 1 mg $L^{-1}$, $N_2$ gas aeration was carried out every 2 days. On the 1st day, 10th day, 20th day, and 30th day from the start of the experiment, three tubes were collected and the RP concentration in the water was determined. The release rate (mg $m^{-2}$ $day^{-1}$) was calculated by dividing the amount of RP in the water by incubation days and by the surface area of sediment samples in the tube (i.e., $1.26 \times 10^{-3}$ $m^2$) for each sample.

### 2.4. Statistical Analyses

To compare the P concentration between the periods of hypoxia (DO $\leq$ 3.0 mg $L^{-1}$) and non-hypoxia (DO > 3.0 mg $L^{-1}$), we performed the Welch's *t*-test or the Wilcoxon signed-rank test using the R (4.3.2) package exact rank test after an F-test to see homogeneity or heterogeneity variance.

## 3. Results

### 3.1. Field Observation

The daily mean water temperature difference between 0.5 m and 5.0 m was largest on 3 August (8.73 °C) and smallest on 21 September 2019 (0.00 °C) (Figure 1b); the daily

mean water density difference was also largest on 3 August (2.12 kg m$^{-3}$) but was smallest on 20 December (0.00 kg m$^{-3}$) in 2019 (Figure 1c). From mid-May to late September 2019, when water temperature and density were largely different between 0 m and 5 m, the DO concentration was low in the bottom layers (5 m and directly above the bottom; Figure 1a). In the present study, the period when DO concentration directly above the bottom was 3.0 mg L$^{-1}$ or less is defined as the hypoxic period. The ORP results showed reduced conditions directly above the bottom during the hypoxic period; the ORP value directly above the bottom was lower than values at 0 m and 5 m during the entire period (Figure 1d).

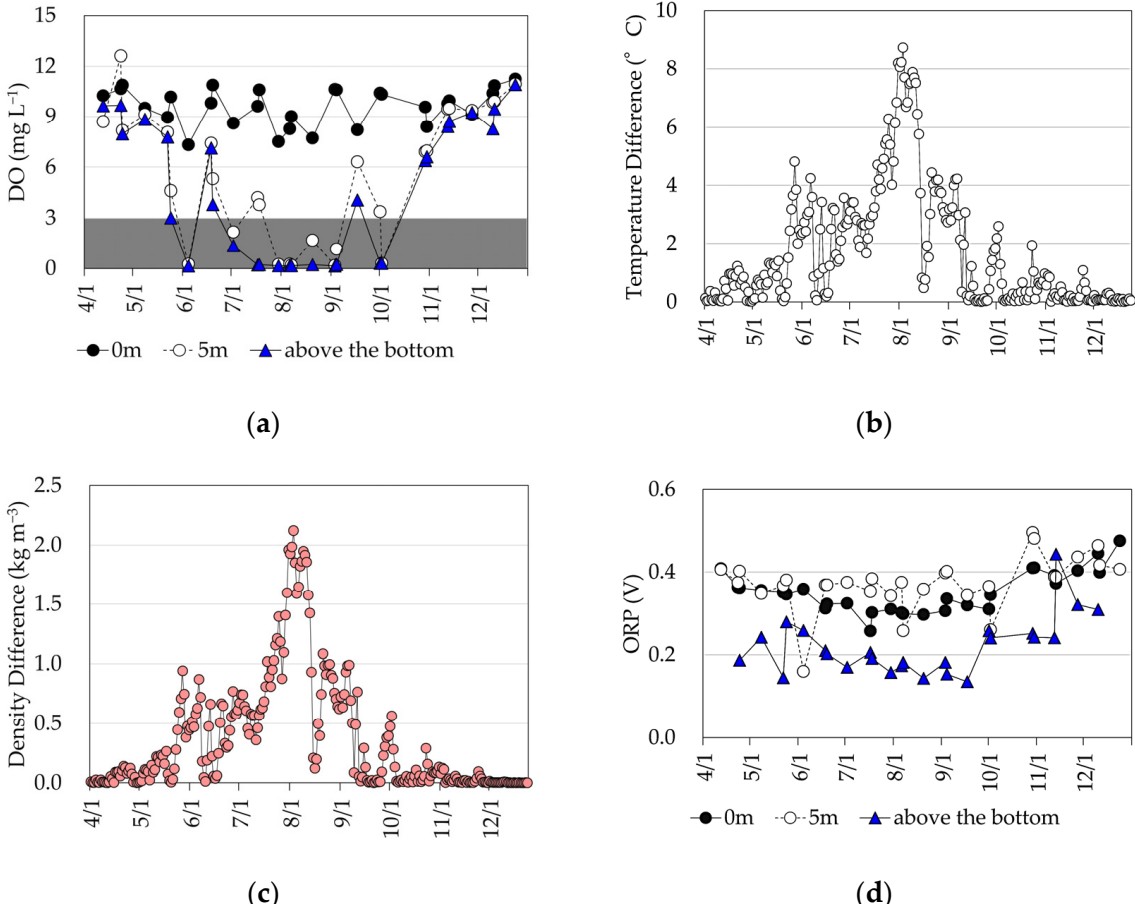

**Figure 1.** Seasonal variation in dissolved oxygen (DO) concentration (**a**); water temperature/density difference between 0.5 m and 5.0 m (**b**,**c**); and oxidation–reduction potential (ORP) (**d**) at Lake Suwa in 2019. Water temperature data measured at 10 min intervals were averaged and expressed as daily data. The gray area in (**a**) indicates hypoxic conditions (DO ≤ 3.0 mg L$^{-1}$).

Whereas there was no significant difference in lake water TP concentration between the hypoxic and non-hypoxic periods at each depth, the concentration was higher at 5 m than at 0 m during the hypoxic period in 2019 (Figure 2b). The RP concentration was significantly higher in the hypoxic period than in the non-hypoxic period at 5 m (Figure 2c). Although not significantly different, the RP concentration in the hypoxic period was higher at 5 m than at 0 m (Figure 2c). The TFe concentration was significantly higher in the hypoxic than in the non-hypoxic period at 5 m, and also significantly higher at 5 m than at 0 m in the hypoxic period in 2019 (Figure 2d).

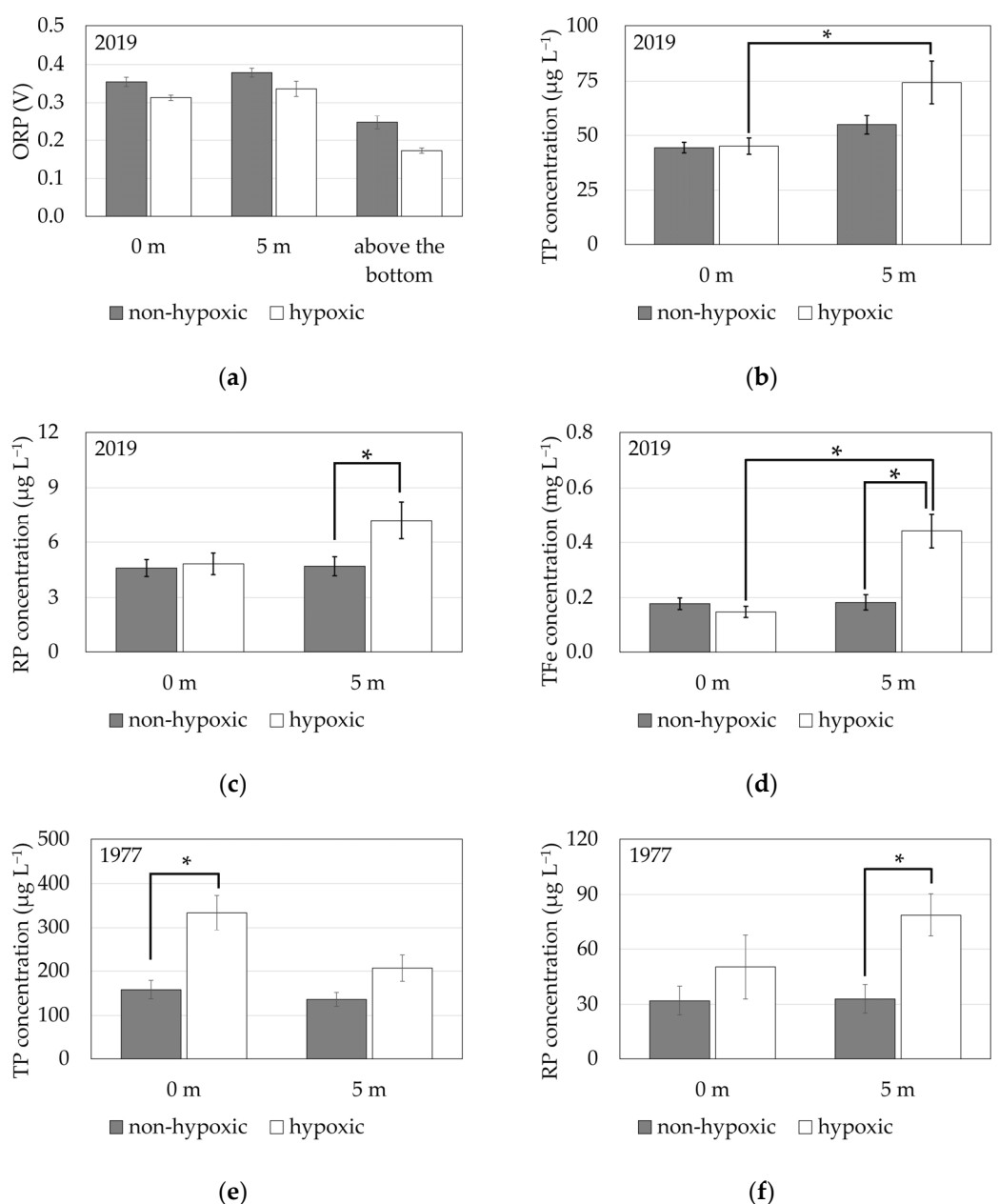

**Figure 2.** Comparison between hypoxic and non-hypoxic periods of oxidation–reduction potential (ORP) (**a**) (non-hypoxia, $n = 15$; hypoxia, $n = 12$); total P (TP) (**b**); reactive P (RP) (**c**); and total Fe (TFe) (**d**) at each depth in 2019 (non-hypoxia, $n = 10$; hypoxia, $n = 8$ at both 0 m and 5 m; non-hypoxia, $n = 7$; hypoxia, $n = 7$ directly above the bottom); and TP (**e**) and RP (**f**) at each depth in 1977 [17] (non-hypoxia, $n = 18$; hypoxia, $n = 4$). Error bar shows standard error. *: $p < 0.05$. Note different scales in (**b,c,e,f**).

The TP content in the sediment remained at approximately 2000 µg g$^{-1}$ during 2019 (1998 ± 27 µg g$^{-1}$, $n = 14$). Among the P fractions examined, only CDB-Pi showed a significantly low concentration in the hypoxic period compared with that in the non-hypoxic period ($p < 0.05$; Figure 3a). The RP concentration in sediment pore water was significantly higher in the hypoxic period than in the non-hypoxic period (Figure 3b). There were no significant vertical differences in TP and TFe content in sediment samples collected in 2020 (Figure 4b), with mean TP and TFe contents being 1961 ± 31 µg g$^{-1}$ and 32.4 ± 0.5 mg g$^{-1}$, respectively.

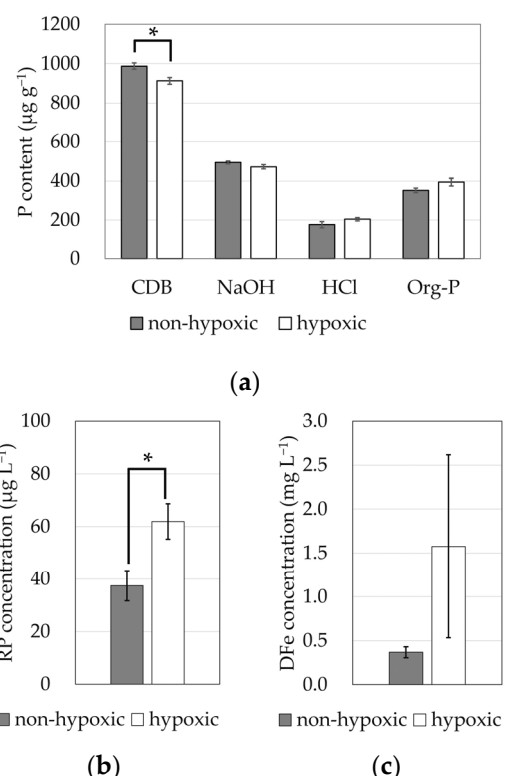

(**a**)

(**b**)                (**c**)

**Figure 3.** Comparison between hypoxic and non-hypoxic periods of P concentration in each fraction in sediments (**a**); reactive P (RP) (**b**); and dissolved Fe (DFe) (**c**) in pore water in 2019. Error bar shows standard error (non-hypoxia, $n = 7$; hypoxia, $n = 7$. *: $p < 0.05$). Note different vertical scales.

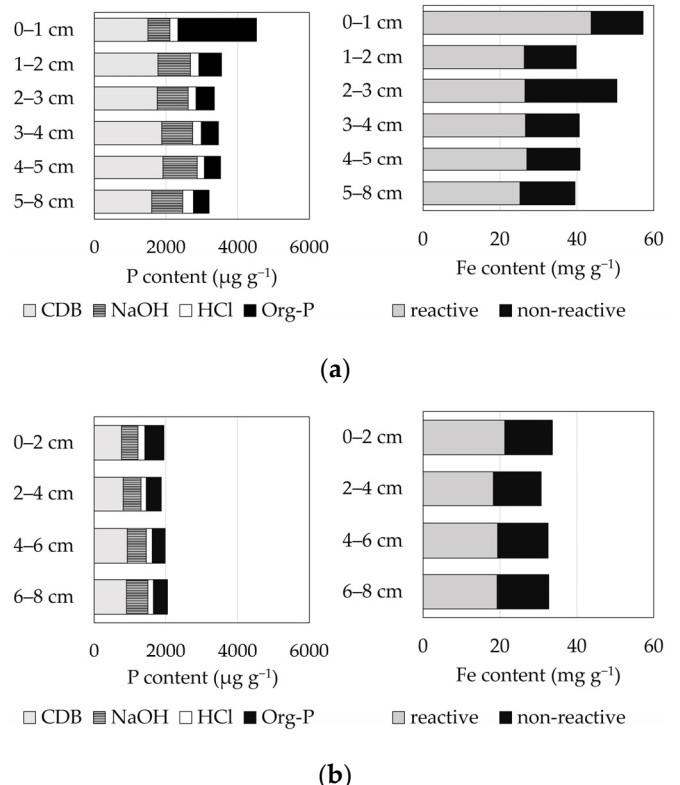

(**a**)

(**b**)

**Figure 4.** Vertical distribution of P and Fe in each fraction in sediment collected in 1977 ($n = 3$) (**a**); and 2009 ($n = 3$) (**b**).

The TP content of settling particles (2167 ± 81 µg g$^{-1}$ at 3 m and 2012 ± 74 µg g$^{-1}$ at 5 m; Figure 5a) was similar to that in the lake sediment (1998 ± 27 µg g$^{-1}$); however, during hypoxic periods (from 18 June to 4 September), the TP content was significantly higher in settling particles than in sediment. The TP content of settling particles collected in the hypoxic periods (2972 ± 140 µg g$^{-1}$ at 3 m and 2843 ± 235 µg g$^{-1}$ at 5 m) indicated values of approximately 1.5 times that of the sediment, due to the remarkable increase in Org-P and CDB-Pi in settling particles during the hypoxic period (Figure 6). Similar to P, the reactive Fe content in settling particles increased during the hypoxic period compared with the non-hypoxic period (Figure 5b). Both the TP and reactive Fe content of settling particles was similar between 3 m and 5 m (Figure 5a,b), but their sedimentation rates were higher at 5 m than at 3 m (Figure 5c,d).

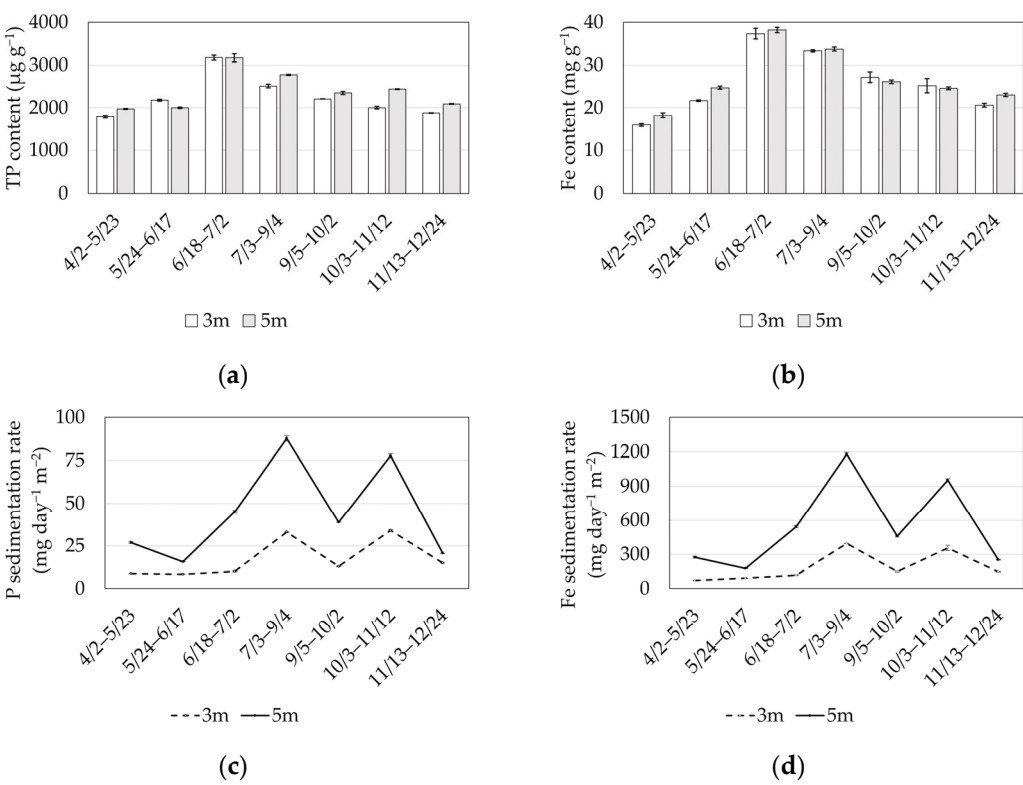

**Figure 5.** Concentration of total P (TP) (**a**) and reactive Fe (**b**) in settling particles, and sedimentation rate of P and Fe at each depth (**c**,**d**) in 2019 (*n* = 3, error bar shows standard error).

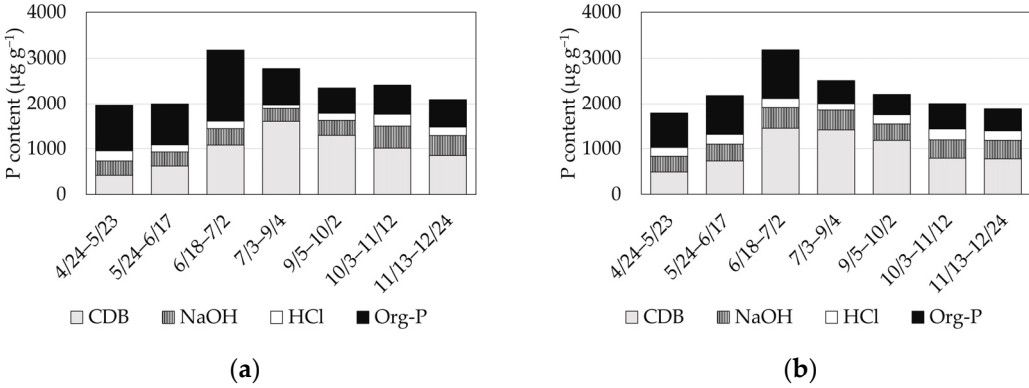

**Figure 6.** Distribution of P among fractions in settling particles collected at 3 m (**a**) and 5 m (**b**) in 2019 (*n* = 3).

The concentrations of suspended P and Fe were calculated by subtracting the concentration of the dissolved form from the total concentration, and were expressed as concentrations of P and Fe in the suspended particles. The suspended P concentration at 0 m showed no significant difference ($p > 0.05$) between the non-hypoxic and hypoxic periods (non-hypoxic period, $3808 \pm 139$ µg g$^{-1}$; hypoxic period, $3976 \pm 361$ µg g$^{-1}$), but the concentration at 5 m was significantly higher ($p < 0.05$) in the hypoxic period ($5507 \pm 406$ µg g$^{-1}$) than in the non-hypoxic period ($4187 \pm 181$ µg g$^{-1}$). A similar trend was also observed for Fe, with no significant difference at 0 m between the periods (non-hypoxic period, $18.6 \pm 1.9$ mg g$^{-1}$; hypoxic period, $16.8 \pm 1.8$ mg g$^{-1}$), but a significant increase at 5 m in the hypoxic period ($37.2 \pm 3.7$ mg g$^{-1}$) from the non-hypoxic period ($15.8 \pm 1.3$ mg g$^{-1}$) ($p < 0.01$). The significantly higher concentrations of both suspended P and Fe at 5 m in the hypoxic period than in the non-hypoxic period were similar to the results found in the settling particles, as described above.

*3.2. Phosphorus Release Test*

The P release test showed a linear increase in water RP concentration with an incubation time from $3.8 \pm 0.2$ µg L$^{-1}$ at day 0 to $361.1 \pm 32.8$ µg L$^{-1}$ after a 30-day incubation, indicating that P was released at 3.7 mg m$^{-2}$ day$^{-1}$ from the sediment in the hypoxic environment.

**4. Discussion**

In shallow eutrophic lakes like Lake Suwa, P concentration throughout the lake water often increases, especially during the hypoxic period, because of internal P loading from the sediment [26]. This P release from the sediment can be caused by several different mechanisms; for example, by reductive dissolution Fe (hydr)oxides, on which P is adsorbed under conditions of hypoxia [27,28]. Recently, however, even in the hypolimnion, there has been no significant difference in the TP concentration in Lake Suwa water between the hypoxic and non-hypoxic periods (Figure 2b), although the RP concentration at 5 m was significantly higher in the hypoxic period than in the non-hypoxic period (Figure 2c). In the epilimnion, TP and RP concentrations were comparable between the two periods (Figure 2b,c). However, during the 1970s–1990s, the P concentration increased not only in the hypolimnion but also in the epilimnion during the hypoxic period, as described below.

Data of lake water P concentrations from 1977 to 2020 can be divided, based on the year 2000 (i.e., 1977–1996 versus 2003–2020), when the external P loading to Lake Suwa declined considerably [29] and, for each data set, TP concentrations were compared between depth and between month (Figure 7a,b). Lake water TP concentration was much higher in 1977–1996 (Figure 7a) than in 2003–2020 (Figure 7b), reflecting the higher amount of external P loading from rivers in 1977–1996 [29]. In 1977–1996, the TP concentration in lake water increased not only in the hypolimnion but also in the epilimnion during the summer; whereas, in 2003–2020, the TP concentration in the epilimnion remained low, even during summer, and increased only in the hypolimnion. There are two possible explanations for the difference in seasonal change of the lake water TP concentration between 1977–1996 and 2003–2020.

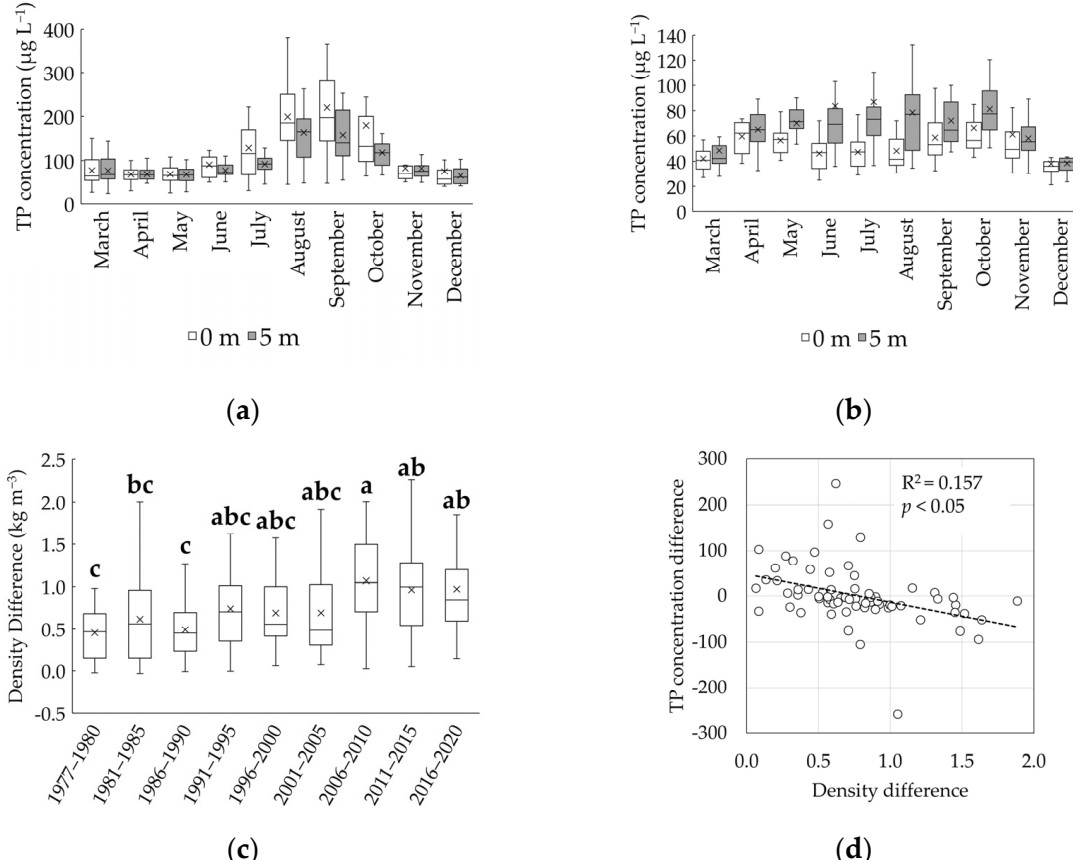

**Figure 7.** Long-term change in water density difference and P concentration of Lake Suwa. Seasonal change in average TP concentration in the past (1977–1996) at 0 m and 5 m depth (*n* = 15–18) (**a**); and in recent years (2003–2020) (*n* = 15–17) (**b**); water density difference (*n* = 16–42) between 0 m and 5 m in summer (July and August) calculated from water temperature (**c**); relationship between water density difference and total P (TP) concentration difference in summer (TP concentration difference was calculated by subtracting the concentration at 5 m from that at 0 m) (**d**). Note that vertical scales differ between (**a,b**). Filled and non-filled bars show the values at depths of 5 m and 0 m, respectively, in (**a,b**). Data are from previous research on Lake Suwa [17,18,30–32], and [unpublished data for 2017–2020]. In the box plot, crosses indicate the average values; vertical bars represent the range; horizontal bars correspond to the median; and the horizontal boundaries of the boxes show the first and third quartiles. In (**c**), mean values without the same letter are significantly different using Tukey's test (*p* < 0.05).

First, the difference in strength of water stratification between 1977–1996 and 2003–2020 may be related to the difference in seasonal changes in the lake water TP concentration between the two periods. The interannual variation in the water density gradient between the hypolimnion and epilimnion during the summer (July and August) showed a significant increase in water density difference with time (r = 0.83, *p* < 0.01), suggesting greater stratification in summer in recent years (Figure 7c). In 2019, there were large differences in water temperature between the epilimnion and hypolimnion from May to September, and the temperature difference—especially from July to August—exceeded the threshold of the water temperature gradient, 1 °C m$^{-1}$ [33], causing stratification. Furthermore, the TP concentration difference between 0 m and 5 m was significantly negatively correlated with the water density difference between 0 m and 5 m in summer (*p* < 0.05; Figure 7d). This suggested that P tended to accumulate more in the hypolimnion with greater stratification. These results imply that internal P loading from the sediment to the epilimnion has been suppressed recently, as a result of the stronger water stratification, compared with the 1970s. A recent increase in air temperature may contribute to the current strengthening of water

stratification in Lake Suwa during the summer (i.e., an increase in the difference in the water temperature and density gradients between the epilimnion and the hypolimnion). According to Jane et al. [34], the strengthening of water stratification in lakes is a recent global trend, which may primarily be caused by global warming. Also, around Lake Suwa, the average summer temperature has increased by 0.41 °C per decade from 1981 to 2012 [35].

Second, the lower P release rate from sediment in 2003–2020 compared with that in 1977–1996, and the readsorption of released P from the sediment on Fe (hydr)oxides (which were reprecipitated via oxidation of Fe(II) to Fe(III) in oxidized water), might contribute to the low P concentrations in the epilimnion even during summers in 2003–2020. The P release rate from sediment in 2022 (3.7 mg m$^{-2}$ day$^{-1}$) estimated in this study was much lower than that in 1977–1979 (13.8 ± 4.1 mg m$^{-2}$ day$^{-1}$ [16]), indicating a lower P release potential of the sediment in recent years compared with that in the 1970s. This is also supported by the lower TP concentration in sediment in 2020 than in 1977: the TP concentration decreased by 44% from 1977 to 2020 (Figure 4a,b). The amount of annual internal P loading in Lake Suwa is estimated to be 13 kg day$^{-1}$, assuming that P release occurs at the release rate we obtained (3.7 mg m$^{-2}$ day$^{-1}$) from the entire lake sediment during the hypoxic period (105 days) in 2022, which was about 19% of the internal P loading in the 1970s (70 kg day$^{-1}$ [16]). Unfortunately, there were no data available on ORP in the 1970s in Lake Suwa, but a comparison of ORP between depths indicated that there was a reduced environment just above the bottom, and that the upper level (i.e., at 0 m and 5 m) has been a relatively oxidized environment in Lake Suwa in recent years (Figures 1d and 2a). Also, in the hypoxic period (June to August), when the lake became a more reduced environment, CDB-Pi values and reactive Fe concentrations in settling particles collected at 5 m were greater than those in the non-hypoxic period (Figures 5b and 6b), and the sedimentation rates of P and Fe at 5 m were higher than those at 3 m (Figure 5c,d). These results indicated that settling particles formed around 5 m during the hypoxic period in Lake Suwa. They also suggest that an 'Fe–P cycle' occurs during the hypoxic period in summer in Lake Suwa recently (Figure 8b): Fe and P are released from sediment through reductive dissolution of Fe (hydr)oxides on which P is adsorbed during the hypoxic period. The released Fe(II) is then oxidized and precipitated as Fe(III) particles, such as Fe oxides, at a depth of approximately 5 m, because the reductive condition just above the sediment gradually changes to an oxidative condition at a depth of approximately 5 m (Figure 1b). Iron(III) particles adsorb released P and then settle to the sediment, preventing supply of released P from the sediment to the surface water. This P adsorption by oxidized and precipitated Fe particles settling to the sediment suppresses the transfer of P from the sediment to lake water. This may reduce the P concentration in lake water, especially in the epilimnion [36].

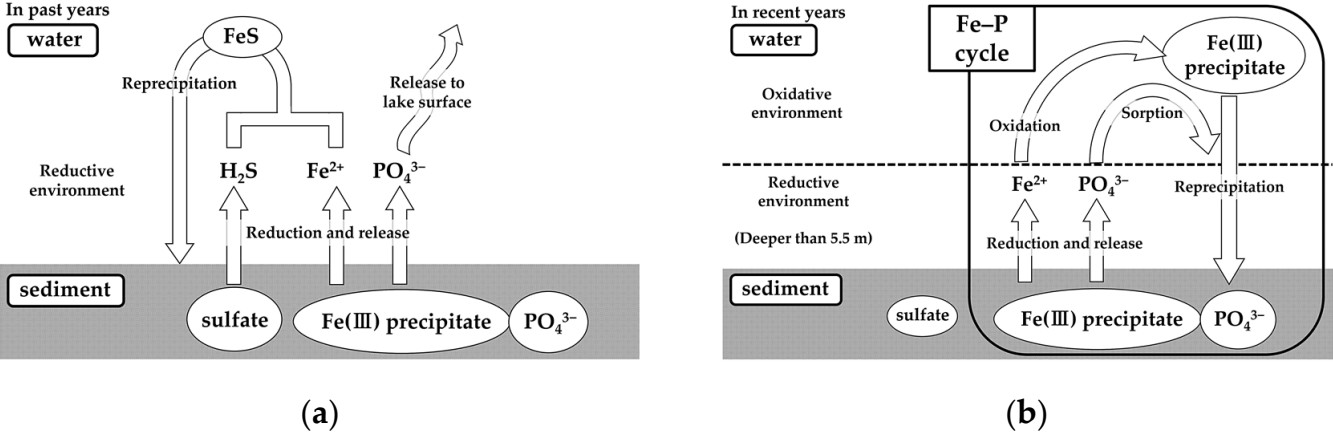

**Figure 8.** Dynamics of P, Fe, and S at the bottom layer during past (**a**) and recent (**b**) summers in Lake Suwa.

The TP concentration in the sediment declined drastically from 1977 to 2020, and so did the Fe concentration to a lesser extent (Figure 4). This was because P was mainly derived from anthropogenic origins and a large amount of P flowed into Lake Suwa through rivers in the 1970s [29], whereas Fe was mainly from natural origins [37]. Accordingly, the ratio of Fe to P in the sediment has increased by about 40% from 1977 to 2020, implying that the P adsorption capacity of Fe in the sediment is greater at present than in the 1970s. Consideration of both points suggests that the amount of P released from the sediments is smaller and, even if P is released, greater water stratification and the stronger Fe–P cycle of the present than in the 1970s (Figure 8a)—caused by the increasing ratio of Fe to P in the sediment—has prevented P supply to the epilimnion in recent years.

It is noteworthy that the sediment TP concentration in the surface layer (0–1 cm) was much higher than that in the deeper layer (1–8 cm) in 1977, although there was no significant vertical change in TP concentration in 2020 (Figure 4). A similar result was also reported for sediments collected in 1979 [38]. This high TP concentration in the surface layer of the sediment was due to the high Org-P concentration (Figure 4a). Hence, this implies that a large amount of Org-P was included in phytoplankton, such as *Microcystis* spp., which bloomed in the 1970s owing to eutrophication [39] and sedimented in the late 1970s. Similar to P, the Fe concentration, primarily as a redox-sensitive Fe fraction, was also high in the surface layer of the sediment collected in 1977 (Figure 4a).

Some factors interrupt the Fe–P cycle (Figure 8a) that prevents P supply from the sediment to the surface water. These include $H_2S$, formed when the sediment is reduced. In the past, when a large amount of decomposed organic matter was present, sulfate was reduced in the sediment and a lot of FeS precipitates were formed, which accumulated in the surface sediment and increased both the total Fe and S concentrations (Figure 8a). At the same time, the sulfide blocked Fe, so that P could be easily released to the lake water [40,41]. However, interruption of the Fe–P cycle by FeS production might be lesser at present than in the 1970s. According to Terashima et al. [37] and the Nagano Prefecture Environmental Conservation Research Institute (unpublished data), S concentration in sediment was found in 2020 to have decreased by 60% compared with 1987, mainly due to sewerage development. A recent decrease in the S concentration in sediment may also suppress P release from sediment to water.

Kawano et al. [42] pointed out the significant role of sediment in the improvement in eutrophication in Lake Suwa. Recently, the quality of the sediment has been successfully improved by decreasing the concentrations of P and S, and increasing the ratio of Fe to P, as described above. This may result from the burial of older sediment with high P and S concentrations by recent sediment with low P and S concentrations due to sewerage development ([37] and the Nagano Prefecture Environmental Conservation Research Institute, unpublished data), with a high sedimentation rate [43]. Terashima et al. [43] and Kanai et al. [44] reported that the sedimentation rate of Lake Suwa was $12.0 \pm 4.1$ mm year$^{-1}$ and 13–14 mm year$^{-1}$, respectively. These values were greater than those of other lakes in Japan: 0–4.1 mm year$^{-1}$ at Lake Kasumigaura [45]; 1–3 mm year$^{-1}$ at Lake Biwa [46]; 1.6–4.8 mm year$^{-1}$ at Lake Hamana [47]; and 4–11.6 mm year$^{-1}$ at Lake Kizaki [48].

Figure 9 shows a schematic diagram of water and sediment quality improvement in Lake Suwa. The amount of external P loading has greatly decreased due to the widespread use of a sewage system. The sewerage development also decreased S loading, and suppressed the promotion of P release from the sediment into lake water by strengthening the Fe–P cycle through a decline in FeS production. In addition, because of the high sedimentation rate, recent sediment with low P and S concentrations (resulting from sewerage development) quickly buried older sediment with high P and S concentrations, resulting in improvement in the surface sediment quality. Furthermore, suppression of lake water mixing by strong water stratification in summer, a consequence of global warming, has constrained internal P loading from the sediment to the epilimnion [36]. In other words, and despite Lake Suwa being a shallow lake, both the reduction in external P loading due to sewerage development and the strengthening of water stratification due to global warming

has suppressed the effects of internal P loading, which has led to a rapid improvement in water quality. As a result, a negative feedback loop has been set up: the reduction in water P concentration reduces algal blooms, which suppresses the supply of organic matter to the sediment. Indeed, Futatsugi et al. [39] reported that the RP concentration in the water of Lake Suwa fell below approximately 10 µg L$^{-1}$ during the period 1999 to 2007, and at the same time the amount of *Microcystis* spp. certainly decreased rapidly. The low level of organic matter content led to less reduced sediment and, consequently, low levels of P supply from sediment to water (Figure 9). Therefore, Lake Suwa is expected to become a more oligotrophic lake in the future. However, since 2005, water chestnut (*Trapa* sp.), a floating-leaved plant, has dominated in the coastal area in summer, covering nearly 20% of the lake surface [49]. A luxuriant growth of *Trapa* sp. may enhance the release of nutrients from the sediment to lake water by decreasing the water DO concentration and by the decay of plant remains [50]; this may result in increased internal P loading and inhibit the negative feedback loop described above. Thus, it is necessary to pay attention to changes in nutrient dynamics caused by thriving *Trapa* sp. for future water quality management in Lake Suwa.

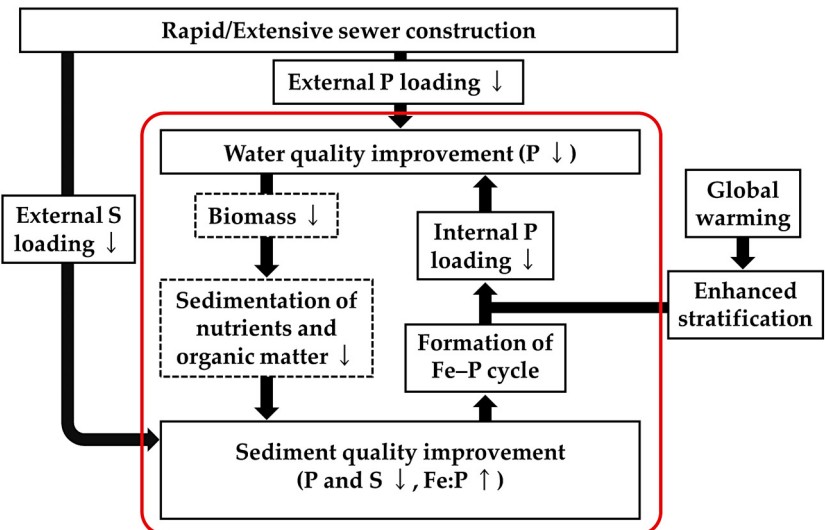

**Figure 9.** Schematic diagram for improving water/sediment quality in Lake Suwa. Negative feedback loop is surrounded by rounded rectangle in red.

Figure 9 shows a schematic diagram indicating the primary importance of effecting a sufficiently reduced P content throughout a lake, by reducing external P loading and so improving the water quality of shallow lakes. A second point is that reducing the internal P loading by strengthening the Fe–P cycle is vital. In addition, efforts to reduce the supply of organic matter to the sediment are necessary, because lowering the organic matter content leads to less reduction in the sediment, so leading to decreased P release from sediment to water. In clarifying the mechanism of water quality improvement in shallow Lake Suwa, the results obtained in this study will provide useful information for improving water quality in other shallow lakes.

**Author Contributions:** Conceptualization, Y.I. and Y.M.; formal analysis, Y.I.; investigation, Y.I.; writing—original draft preparation, Y.I. and Y.M.; writing—review and editing, T.K. and Y.M.; visualization, Y.I.; funding acquisition, Y.I. All authors have read and agreed to the published version of the manuscript.

**Funding:** This work was supported by a JSPS Grant-in-Aid for JSPS Fellows, Grant Number JP 21J21859.

**Data Availability Statement:** Data obtained in this study are contained within the article. Earlier data sources are all given in Figure 7 legends.

**Conflicts of Interest:** The authors declare no conflicts of interest.

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
