# Peer review of "Temporal Phosphorus Dynamics in Shallow Eutrophic Lake Suwa, Japan"

_water, doi:10.3390/w16101340_

Round 1

Reviewer 1 Report

Comments and Suggestions for Authors

In this study, the author determined the sedimentary P species in the sediments and suspended particles, RP in the water, and P release from the sediments in the Lake Suwa, Japan. The work is comprehensive and the paper is well written, it is easy to read and comprehend. However, I do have some minor comments.

Materials and Methods:

- Explain which depth layers are considered epilimnion and hypolimnion. Are the 0 m and 5 m depths serve as representatives of epilimnion and hypolimnion, respectively? If so, explain accordingly.

- Figure 1: what is meant by above the bottom? Do you mean sediment intersitial water? The author mentioned 0 m and 5 m depth of water, but there was no mention of above the bottom water in the Method section.

- Figure 2: (c) and (f) are the same.

Results:

Lines 205-206: Do you mean the concentrations of TP and TFe in the suspended particles werecalculated by subtracting their concentrations in the dissolved form from their total concentrations in the sediments?

Discussion:

Figure 7: explain what are the filled and non-filled bars in (a) and (b) represent.

Lines 294-298: How did the author infer that these CDB-Pi and Fe were of settling particles and not release from the sediments?

Author Response

We appreciate the comments and suggestions on our manususcript.

Reviewer 2 Report

Comments and Suggestions for Authors

The manuscript presents the study on the phosporus content and distribution in Lake Suwa, Japan. Being a shallow lake situated in the urban area, Lake Suwa is experiencing a heavy anthropogenic load. Nevertheless, due to a number of preservation initiatives as well as to the climatic changes, the phosphorus load in the lake is decreasing during the last decade. The authors show the dynamics of this process starting from 1970s and reveal its mechanisms.

There are several points that should be addressed.

L. 46: What are the reasons why Lake Søbygaard is taken to compare with Lke Suwa?

L. 44-45, 48: It would be more convenient to compare the numbers of external phosphorus loading in two lakes if these numbers are expressed in the same units. Is it possible to recalculate (probably using the area of the lake)?

L. 145-147: Please add the year directly to the month (e.g. 3 August 2019, From mid-May to late September 2019)

Fig. 1a, d: The highlighting of the hypoxic zone below 3 mg/L (a) and of the reduced conditions (d) will make the figures easier to read.

Fig. 2: In the Materials and Methods section it is indicated that the samples were collected from the lake surface (0 m) and near bottom (5 m) (L. 92). At the same time, the text in the L. 160-161 is "concentration was higher at 5 m and directly above the bottom than at 0 m". Please comment what was the depth directly above the bottom in this case.

Fig. 4: Probably it would make more sence if the data of 1977 come first (a), and then the data of 2020 (b). The same for Fig. 8.

Fig. 6, L. 203-204, Fig. 7, L. 255-256: The letter designations on the bars (a-f) are not clear. Please specify.

Little is said about the phytoplankton blooms in the lake. It would be interesting to see the comparison between the frequency and intensity of the blooms during the last decade, taking into account that, as it is said in L. 371, the reducing P concentration also reduced the blooms.

Author Response

We appreciate the comments and suggestions on our manuscript.

Reviewer 3 Report

Comments and Suggestions for Authors

The article describes the P dynamics in the shallow, weakly stratified Lake Suwa, where the phosphorus load has decreased many times over in the last 50 years. A comparison of data on P concentrations in water, sediment and settling particles as well as data on the stability of stratification in the 1990s and in 2019 clearly shows the differences between the P cycle in both periods. The article is clear, logical and formally very precise.

The only minor comment I have is that the authors did not compare the phosphorus loading to the lake with the amount of P retained in the sediment in the assessment, which could have illustrated the differences between the two time periods even better than their approach. If the authors choose not to add this, I do not see it as a hindrance to publication, but I would like to see more detailed hydrologic data on Lake Suwa included in the methodology, showing precipitation data, temperature data, flow data, and residence time of water in the reservoir.

Another remark is that 55% of the referenced literature is in Japanese and therefore difficult to access for an international audience. Perhaps equivalents from international journals could be used instead of some Japanese papers, e.g. for methodological reference [21].

Detailed comments:

l. 82: Please include data on: 1) average precipitation, preferably for both the catchment area and the lake area, but at least for the Lake Suwa area [e.g. in mm/year]; 2) long-term average water inflow into the lake; 3) water temperatures in the lake, i.e. temperature during mixing from autumn to spring and surface temperature in summer, etc.

l. 90: Change "min" to "minutes".

l. 95: Delete without replacement the word string "the filtrate water from".

l. 330-333: The connection between the loss of S in the sediment and the development of sewerage is correct, but superficially described. I think that the decrease in S concentration in the sediment is not the direct cause of the suppression of P release from the sediment. As shown in Fig. 8, in the past, when a large amount of decomposed organic matter was present, sulphate was reduced in the sediment and a lot of FeS precipitates were formed, which accumulated in the sediment and increased both the total Fe and S concentrations in the sediment. At the same time, the sulphide blocked Fe so that P could be easily released. At present, the Fe content in the sediment is lower because FeS is apparently no longer formed, but the Fe is more effective in precipitating P and does not allow it to be released from the sediment as much as it was in the past.

It would be helpful if you could describe this mechanism instead of the sentence in lines 311-333.

End of comments

Comments on the Quality of English Language

The only linguistic comment I have is the use of the term "elution", i.e. "P elution from sediment" (line 54, line 55) and "The elution rate" (line 134). In the context of the release of substances from the sediment, the use of the term "elution" is inappropriate, as it is originally used for "the process of extracting one material from another by washing with a solvent to remove adsorbed material from an adsorbent" (e.g. in Wikipedia), which is usually not the case for diffusion processes during the transport of substances at the interface between sediment and water column. In all cases, I recommend replacing the term "elution" with the simple and technically neutral "release".
End of comments

Author Response

(The authors gave the same response as above.)
